# Dissecting eustachian tube dysfunction: From phenotypes to endotypes

**Cuneyt M. Alper**[1,2]\*, **Miriam S. Teixeira**[3], **Ellen M. Mandel**[1], **J. Douglas Swarts**[1]

**1** Department of Otolaryngology, University of Pittsburgh School of Medicine, Pittsburgh, PA, United States of America, **2** Division of Pediatric Otolaryngology, UPMC Children's Hospital of Pittsburgh, Pittsburgh, PA, United States of America, **3** Graduate Medical Education Research Division, Arnot Ogden Medical Center, Elmira, New York, United States of America

\* cuneyt.alper@chp.edu

## Abstract

### Objective

A broad spectrum of complaints, symptoms and manifestations has been assigned to Eustachian tube (ET) dysfunction (ETD). While such presentations may manifest as ETD phenotypes, underlying mechanisms are defined as endotypes. Our goal is to develop a diagnostic approach to differentiate the endotypes and guide clinicians in the workup and selection of treatments targeting the mechanism of ETD.

### Study design

Retrospective.

### Setting

Tertiary care.

### Subjects and methods

Children and adults with suspected ETD were evaluated with a thorough examination, otomicroscopy, otoendoscopy, trans-nasal videoendoscopy and testing of passive and active ET dilatory properties. Degree of weakness in soft palate elevation and ET orifice widening (muscular weakness, ETD-M), presence of inflammation (ETD-I) and/or adenoid tissue impinging and restricting the ET opening (ETD-R) were assessed with video-endoscopy. The Forced Response Test, Inflation-Deflation Test and Pressure Chamber Test were used as applicable to quantify the degree and type of difficulty (Stricture, ETD-S or adhesive, ETD-A) or ease (patulous or semi-patulous, ETD-P/SP) in opening the ET, and degree of active muscular strength/weakness (ETD-M) was measured. Ears with normal function (ETF-N) findings were also identified.

### Results

Video-endoscopic and ETF test results were obtained for 71 ears of 40 subjects (22 males, 18 females; 38 white, 2 black), with an average age of 22.9 ± 16.5 years (min:6.2,

**Data Availability Statement:** The data available from the University of Pittsburgh Institutional Data Access. The data underlying the results presented in the study will be made available by the corresponding author.

**Funding:** CMA, MST, EMM, DS (PI William J, Doyle, later CMA) received the award from National Institute of Health, National Institute on Deafness and Other Communication Disorders Grant DC007667. The funders had no role in study design, data collection and analysis, decision to publish, or preparation of the manuscript.

**Competing interests:** No authors have competing interest.

max:64.1). Videoendoscopy (21, 13, 33, 16, 13, 0, 0 ETs) and ETF testing analysis (20, 24, 0, 38, 0, 3, 13 ears) were categorized as ETF-N and the ETD endotypes ETD-S, ETD-R, ETD-M, ETD-I, ETD-A, and ETD-P/SP, respectively. Some phenotypes had features consistent with more than one endotype.

## Conclusion

A systematic approach of examination and testing may differentiate the specific underlying mechanisms, lead to a treatment targeted to the ETD endotype and may establish novel ways to diagnose and treat ETD.

## Introduction

Otitis Media (OM) is a disease characterized by inflammation of the middle ear (ME) mucosa with or without the accumulation of ME effusion (MEE) in the normally air-filled ME [1]. OM is common in the pediatric population but also occurs in older children and adults, albeit at lesser prevalences [2–4]. The cost of diagnosing and treating OM in the US is greater than $5 billion/year [5].

There is a broad spectrum of symptoms, complaints or manifestations that is traditionally attributed to Eustachian tube dysfunction (ETD) including: recurrent or chronic otalgia, ear pressure, autophony, otitis media, recurrent acute otitis media, otitis media with effusion, chronic otitis media with effusion, tympanic membrane (TM) retraction, retraction pocket, cholesteatoma, TM perforation, need for or habit of frequent sniffing, barotrauma, diver's otalgia or otitis, flight-related otitis, inability to pop the ear, need for ventilation tube (VT) insertion, failure of TM perforation repair, and recurrence of OM and/or symptoms after tympanoplasty. While many of these may directly be associated with ETD, this commonly used explanation for these complaints/problems does not help understand or resolve the condition unless there is actual evidence that ETD is the cause and there is a method to resolve or mitigate this factor.

Numerous studies support a causal role for ETD in the development and persistence of OM with effusion (OME) [6] and it may facilitate the development of acute otitis media (AOM) and recurrent AOM [7] (RAOM) by providing a low resistance conduit by which nasopharyngeal (NP) bacteria/viruses gain access to the ME [8–10]. Sustained negative pressure secondary to ETD is responsible for TM retraction and retraction pockets (TM-R/RP), MEE and formation of acquired cholesteatoma [11–14]. On the other hand, there is no consensus regarding the definition, criteria, classification or grading of ETD. Moreover, there is no clear understanding regarding the underlying mechanisms or pathogenesis of ETD and whether there are multiple endotypes leading to the outlined manifestations.

The ET is a biological tube connecting the ME to the nasopharynx (NPx) consisting of a posterior osseous tube—an extension of the ME—and an anterior functional membrano-cartilagenous portion that continues to the NPx [15]. The mucosa of the tubal lumen is continuous with that of the ME and NPx and shares with the ME a mucociliary clearance mechanism that propels mucus to the NP [16]. Under normal conditions, the ET is closed by periluminal pressures in excess of ambient, but its lumen is opened periodically during swallowing and other NP activities involving contraction of the Tensor Veli Palatini muscle (mTVP) with perhaps the assistance of the Levator Veli Palatini muscle (mLVP) [17]. The normal ET serves three functions related to the preservation of ME health: protection of the ME from NP pressures

and pathogens (development of RAOM); clearance of ME fluids and debris, and maintenance of MEP at near-ambient levels [7].

Until recently, the only treatment for the consequences of ETD was to temporarily bypass the ET by the insertion of a ventilation tube (VT). This approach does not modify ET function (ETF) itself but rather it buys time until the ETD resolves, or VT re-insertion is necessary to maintain an aerated ME. Lacking a better alternative, the sequelae and complications resulting from VT insertion are considered acceptable. The recent introduction of balloon dilation of the Eustachian tube (BDET) changed that landscape [18–27]. Unfortunately, innovation and advancements in balloon technology were not mirrored by the dissemination of knowledge and experience concerning ETD. Without the tools to identify the patients with ETD, determine which of them would benefit from BDET, and a method to assess and measure its effects, treatment recommendations are based on clinician's opinion.

The simplistic concept that the ET is either "too closed" or "too open" ignores the reality that a spectrum of mechanisms causes ETD, creating a diagnostic and treatment challenge. Conceptually, VT insertion does not solve the underlying mechanism, but temporarily solves the "too closed" ET by bypassing and taking over its pressure equalizing function. Similarly, BDET targets the "too closed" ET by making the ET presumably easier to open. The only evidence in the literature, based on a very limited number samples, attributes how BDET changes ETF to damage of the inflamed mucosa with the applied intraluminal circumferential pressure during the 90–120 second inflation of the balloon, leading to replacement with normal mucosa [24]. However, mucosal inflammation may not necessarily be the only mechanism of ETD, and if BDET helps patients with non-inflamed ETs, it is not wrong to assume the existence of other mechanisms of efficacy.

While ETD phenotypes manifest as a broad spectrum of clinical presentations with a variety of elements of history, symptoms and clinical findings, with the exception of apparent patulous ET, they are rarely diagnostic. A number of such elements may be in the patient's history, or current findings may just be sequelae of the now-resolved ETD. Moreover, variability in ETF over time, with intermittent symptoms and findings due to exposure to seasonal or incidental inflammatory conditions or to ambient pressure variations, i.e., "baro-challenge", limit the individual's ability to identify, differentiate and express his/her complaints in a consistent and reliable way. This manifests as the limited sensitivity and specificity of diagnostic questionnaire tools such as ETDQ-7 [28,29]. BDET has the risk of causing or worsening a patulous ET and the failure of questionnaires to identify this condition is a major concern [30]. Also, inability to differentiate ear-specific ET symptoms reduces the usefulness of such instruments [29]. Even a panel of world experts on ETD failed to diagnose ETD and differentiate it from non-ETD etiologies based on patient-reported symptoms [31]. This study concluded that ETD should be diagnosed on the basis of clinical assessment and tests of ET opening, as patient-reported outcome measures were shown to have no diagnostic value.

Offering BDET for all non-patulous ETD phenotypes assumes either a single common pathogenesis or that balloon dilation works successfully on all potential mechanisms. There is currently limited understanding of such distinct paths of pathogenesis of ETD. Identifying the mechanisms that result in ETD phenotypes is crucial in defining the ETD endotypes. Based on the identified mechanisms and the description of distinct clinical and testing criteria, selection of the best treatment for each endotype will be possible while reducing the risk of worsening ETD with inappropriate treatments. Therefore, a study was conducted employing video-endoscopic and ETF test results of patients evaluated in an ETD specialty clinic to identify the different mechanisms that contribute to the dysfunction of the ET. Our goal was to define specific ETD endotypes based on clinical and testing features and establish a diagnostic pathway to guide clinicians and researchers from phenotypes to mechanisms of ETD and discuss how this approach can be used to better understand ETD and improve treatment outcomes.

## Material and methods

For this study, we analyzed existing video-endoscopic and ETF test results of the first 40 subjects with complete evaluation and testing in the ETD Specialty clinic (ETDC) at the UPMC Children's Hospital of Pittsburgh. Seventy-one ears of 40 children and adults with suspected ETD were evaluated with a thorough history, ENT exam, tympanometry, otomicroscopy, trans-nasal video-endoscopy and a set of ETF tests. Use of de-identified data for analysis was approved by the University of Pittsburgh Institutional Review Board (IRB) reviewed and approved as retrospective chart review without a need for consent, with the IRB (OSIRIS) number PRO14110237. Consent was not obtained from the subjects.

ETF tests were used as applicable for ears with intact and non-intact TMs. The specific tests used to investigate the passive and active ET dilatory characteristics were as follows: 1) ET passive properties for non-intact TMs from the Forced Response Test (FRT), including opening pressure (OP), steady resistance (RS) and closing pressure (CP) and the Valsalva and Sniffing maneuvers; 2) ET active properties from Sonotubometry, FRT dilatory efficiency (DE) and Tubomanometry, and percentage equilibration of positive and negative MEP on the Inflation-Deflation test (IDT) for non-intact TMs and on the Pressure chamber test (PrChT) for intact TMs.

Trans-nasal video-endoscopy was used to assess soft palate elevation and ET orifice widening (muscular strength/weakness) and presence of inflammation and/or adenoid tissue impinging on or restricting the ET opening; the OP, RS, CP and Valsalva were used to quantify the difficulty or ease in passively opening or closing the ET; DE and percentage of MEP equilibration (%MEPEq) to quantify active para-tubal muscular strength; Sonotubometry and Tubomanometry to detect ET opening thresholds and the Sniff maneuver evaluated ET weakness in ME protection.

### Specific methods and interpretation for Eustachian Tube Dysfunction (ETD) endotypes

Specific findings and their interpretation with potentially associated ETD endotypes are summarized in Table 1. The description of the specific methods and their interpretation is outlined below.

**Otoscopy/Otomicroscopy.** An intact TM in neutral position in the absence of ME effusion implies that gas is able to get into the ME through the ET. In the absence of recent forceful insufflation of gas, this condition suggests that the ET is not obstructed, although ETD-P or ETD-SP would not be excluded. The presence of effusion, especially when completely filling the ME, with or without evidence of acute infection, precludes ETF testing. Effusion does not necessarily imply ETD, since it could be strictly infectious, especially in infants and toddlers. Tympanic membrane retraction or retraction pocket (TM-R/RP) is more likely related to ET obstruction; however, it can also be present in ETD-P or ETD-SP. TM movements synchronous with breathing which become more pronounced with forceful breathing confirms ETD-P. However, for ETD-SP, instead of continuous synchronous movements, the TM suddenly pops out and in during more forceful breathing or breathing when one nostril is pinched.

**Tympanometry.** Measurement of MEP is essential in assessing MEP regulation, which is a combined outcome of both ME mucosal gas exchange and the ability of the ET to passively or actively equilibrate pressure differences between the ME and NPx. Tympanometry is also an essential adjunct test method to measure a change in MEP during the other ETF tests. Without verification of a pressure change, it would not be possible to perform or interpret most tests. Tympanometry is typically used in ears with intact TMs and is helpful in

**Table 1. Criteria, findings and interpretation of assessments and test results and possible associated ETD endotypes.**

| ASSESSMENT / TEST | CRITERIA / FINDING | INTERPRETATION | POSSIBLE ENDOTYPE |
|---|---|---|---|
| **Otoscopy** | Non-intact TM | Cannot determine | |
| | TM movement synchronous with breathing | Poor protection | ETD-P, ETD-SP |
| | TM position–neutral | Adequate ventilation | Normal, ETD-P |
| | TM position–retracted | Poor ventilation | ETD-S, -R, -I, -M, -A |
| | Middle ear–clear | Adequate ventilation | Normal, ETD-P /-S /-R /-I /-M /-A |
| | Middle ear–effusion | Inadequate ventilation and/or clearance | ETD-S, -R, -I, -M, -A |
| **Trans Nasal Video-Endoscopy** | Secretions/ edema | Infection /inflammation | ETD-I |
| | Adenoid/ lymphoid tissue—no impingement | Inflammatory reservoir | ETD-I |
| | Adenoid/ lymphoid tissue with impingement | Limited ET widening | ETD-R |
| | Weak soft palate elevation | Inadequate mLVP function | ETD-M |
| | Inadequate/ short velar closure | Velopharyngeal incompetency | ETD-M |
| **Sonotubometry** | Absent sound pressure elevation | Absence of ET opening | ETD-S, -R, -I, -M, -A |
| | Continuous high sound pressure | Continuously open ET lumen | ETD-P |
| **Forced Response Test (FRT)** | High passive features (OP, RS, CP) | Difficult to open ET lumen | ETD-S, -R, -I, -M, -A |
| | Low passive features (OP, RS, CP) | Low protection | ETD-P /SP |
| | Dilatory efficiency > 1 | Widening of ET lumen with swallow | Normal |
| | Dilatory efficiency = 1 (absent) | Inability to widen the ET lumen | ETD-S, -I, -M, -A |
| | Dilatory efficiency < 1 | Narrowing of ET lumen with swallow | ETD-R |
| **Pressure Equilibration Test with IDT or PrChT** | Inability to establish pressure gradient | Low tissue pressures and poor protection | ETD-P, -SP |
| | High % correction of pressure gradient | Normal active muscular function | Normal |
| | No /Low % correction of pressure gradient | Inadequate active muscular function | ETD-M, -S, -R, -I, -M, -A |
| **Tubomanometry** | Concurrent nasal and ear pressure change | C1 = P1, open ET lumen or low protection | ETD-P |
| | Absent R value at all pressures (30–50 mbar) | Difficulty in ET opening | ETD-S, -R, -I, -M, -A |
| | R value >1 at high pressure (50 mbar) | High opening pressures | ETD-S, -R, -I, -M, -A |
| | R value >1 at 30–40 mbar | Normal passive (possible also active) function | Normal |
| | R value <1 at low (30 mbar) pressure | Poor protection | ETD-SP |
| **Valsalva** | Unsuccessful ET opening | Low nasal pressure or high ET opening pressures | ETD-S, -R, -I, -A |
| | High residual pressure | High closing pressure (CP) | ETD-S, -R |
| | Spontaneous neutralization of TM bulge | Low tissue pressures, poor protection | ETD-P /SP |
| | No residual pressure after Valsalva | Low closing pressure, poor protection | ETD-P /SP |
| **Sniffing** | Absent ET opening with forceful sniff | Good ET protection, high tissue pressures | ETD-S, -R, -I, -A |
| | High negative residual pressure (locking) | High closing pressure (CP) | ETD-S, -R |
| | Spontaneous neutralization of TM retraction | Low tissue pressures, poor protection | ETD-P /SP |
| | No residual pressure after sniff | Low closing pressure, poor protection | ETD-P /SP |

Abbreviations = TM: Tympanic membrane; ET: Eustachian tube; ETD: Eustachian tube dysfunction; ETD-P: Patulous; ETD-S: Obstructive -Stricture; ETD-I: Obstructive-inflammatory; ETD-R: Obstructive- Restrictive /Lymphoid/ Adenoid tissue; ETD-A: Obstructive–Adhesive; ETD-M: Active muscular; mLVP: Levator Veli Palatini muscle; OP: Opening pressure; RS: Steady state resistance; CP: Closing pressure; PEqT: Pressure Equilibration Test; IDT: Inflation deflation test; PrChT: Pressure Chamber Test.

diagnosing MEE. When the ME is completely filled with effusion, the tympanogram reveals a Type B (flat) curve. If there is some air in the ME, there may be an identifiable peak with a broad MEP gradient. Regarding ETF, tympanometry itself is not typically too informative. However, a Type A tympanogram, if there was not a forceful opening of the ET earlier with a maneuver like Valsalva, indicates a healthy MEP regulatory function, specifically normal ETF or an easily opening ET, i.e., ETD-P or ETD-SP.

**Sonotubometry.** A sudden increase in sound pressure level transferred through the ET lumen is a method to verify an opening of the ET. A nasal probe delivers the sound at a pre-determined spectrum of sound frequency / white noise and a microphone records the sound that reaches the external ear canals (EEC). While the presence of an increased sound in the EEC when there is no pressure gradient between the NPx and ME indicates an excellent active muscular function or ETD-SP, a continuously high sound pressure level indicates the ETD-P endotype. Absence of an increase in the sound level during swallow when there is no pressure gradient does not indicate an abnormal ETF, but when there is a considerable pressure gradient between the NPx and ME, good active muscular function is expected to facilitate the opening. Sonotubometry can also be useful in real-time detection of ET opening when a pressure gradient is generated via a tympanometer or tubomanometer or inside a pressure chamber, but special multi-purpose probes are needed for those assessments.

**Forced Response Test (FRT).** In a non-intact TM, parameters of ET opening while running air through the EEC-ME-ET have been used to assess the passive and active properties of ET function. A pump delivering a standard flow (typically 11 or 23 ml/min) into a relatively stable (pre-opening of the ET) volume increases the system pressure to a level that exceeds the tissue pressures and opens the ET at the OP. The system pressure decreases to a steady-state pressure (PS) when the flow rate equalizes to the pump flow rate. The difference between the PS and the OP reflects the additional force needed to initially separate the closed surfaces of the ET lumen, most of which may be accounted for by the mucosal adhesion properties, surface tension properties or the lack thereof, the gradual give of the elastic fibers to displace and separate the ET lateral and medial walls. During the PS, the steady-state resistance to airflow (RS) can be calculated, which is a measure of lumen volume. Active muscular function of the ET is measured during this steady state at PS. The effect of a single patient-initiated swallow on pressure and flow parameters is a direct measure of how well an ET can equilibrate a pressure gradient. Typically, a normal ET is expected to widen with active muscle contraction: first, the mLVP elevates the soft palate, rotating the posterior lamina of the ET cartilage, widening the apex angle; second, the mTVP widens the ET lumen. Dilatory efficiency (DE) is calculated as the increase in air flow with active muscular function, reflecting the increase in area of ET lumen.

**Pressure Equilibration Test (PEqT).** The ability of an ET to equilibrate a pressure gradient with a standard physiologic maneuver is a method to assess and quantify the active muscular function of the ET. To have comparable results across different test methods, pressure gradients of +200 daPa and -200 daPa are used as standard starting pressures. The patient is then asked to swallow 5 times, and % correction at 1 and 5 swallows is used as a parameter of active ETF. Typically, equilibration of a positive pressure gradient is easier compared to a negative gradient.

**MEPEq with Inflation Deflation Test (MEPEqT$^{IDT}$).** When TMs are non-intact, pressure in the EEC-ME–Mastoid Air Cell System is modified with an ear canal probe attached to a syringe with a T-connector and a pressure sensor. The sealed system pressure is changed to the desired level and checked for stabilization, and the patient is asked to swallow 5 times at 5-second intervals.

**MEPEq with Pressure Chamber (MEPEqT$^{PCh}$).** When there is an intact TM, a Pressure Chamber is used to create a ME-NP pressure differential of +200 daPa and -200 daPa. The patient is then asked to swallow 5 times, and % correction at 1 and 5 swallows is used as a parameter of active ETF.

**Tubomanometry.** A test device to increase the nasal-NP pressure in a controlled stepwise approach while concurrently monitoring the change in the EEC pressure is the basis of the Tubomanometry device. Despite the weaknesses originating mostly from the software and interpretation of the output, this is a valuable testing instrument for the assessment of ETF. After sealing ear probes in the EEC, a two-pronged nosepiece is sealed into both nostrils and the external flow outlet is manually closed. The person is asked to swallow and the increase in NP pressure triggers the tubomanometer to suddenly release an air-bolus to achieve a pre-set pressure level. With the re-opening of the velum at the end of the swallow, the nasal pressure decreases. In the event of an ET opening, the ear probe either senses an increase in EEC pressure due to the outward movement of the TM or senses the direct increase in MEP if TM is non-intact. The Tubomanometer is set to intact and non-intact TM conditions before the testing which adjusts the displayed ear probe pressure scale accordingly. The described standard test protocol uses a sequence of 30, 40 and 50 mbar nasal pressure levels, and the R value is defined by using a formula derived from the time that there is an ET opening with respect to the phase and time of the Nasal-NP pressure curve and is used to describe ETF. Absence of a change in the EEC pressure yields no R value, indicating that the ET has not opened. ET openings that happen concurrently with the increment in Nasal-NP pressure curve suggest the ETD-P or ETD-SP endotypes. A change in the EEC pressure before the nasal/NP pressure reaches the target pressure results in an R value that is ≤1, indicating an ET opening below the pre-set target system pressure. If an ET opening occurs after the Nasal-NP pressure reaches the target pressure and starts a plateau, the R value becomes >1.

**Valsalva maneuver.** Air is forced out of the lungs against pinched nostrils and a closed mouth. The soft palate remains relaxed and directs the air pressure to both nasal and oral passages, increasing the nasal/NP pressure to a level dictated by the strength of expiratory muscles. It is important to note that even though this is an active maneuver, opening of the ET lumen is a passive process, therefore Valsalva is considered a passive ETF test method. When the pressure gradient between the NPx and ME exceeds the OP, i.e., the tissue pressures, the ET lumen opens and, based on the duration of time that this excess pressure gradient exists, there is a continued column of air through the ET lumen. When the Valsalva maneuver is over and nasal/NP pressures return to ambient pressure, there is a backflow from the over-pressurized ME to the NPx, which is considered the closing pressure (CP). A high residual pressure indicates ETD-S endotype.

**Sniff maneuver.** A rapid and forceful inhalation of air through one nostril decreases the nasal/NP pressure rapidly. If the negative NP-ME gradient exceeds the ET OP and creates a negative MEP, it suggests an ET with low protective characteristics, i.e., a patulous (ETD-P) or semi-patulous (ETD-SP) ET. The low residual pressure, if present, is expected to spontaneously "leak" or easily be cleared after a swallow. An ETD-S endotype is not expected to open the ET with a sniff. In extreme pressure differences, if it does open, a negative residual pressure is expected to remain, and it would be difficult for ETD-S, ETD-A or ETD-M endotypes to clear.

**Trans nasal video-endoscopy.** Visual evaluation of the NPx is useful for documentation of patency of the nasal passages, quality and quantity of nasal secretions, presence and degree of postnasal drip, the degree to which these secretions are coming over or covering the ET orifice, and the degree and severity of appearance of mucosal inflammation in the nasopharynx, specifically in the peri-tubal area, on the Torus Tubarius and into the ET orifice. The presence

of adenoid tissue or residual/recurrent adenoids and their location in the NPx, at the Fossa of Rosenmuller or over the Torus Tubarius is also important information obtained during the endoscopic examination. Both static and dynamic video-endoscopy need to be performed with 0˚ and 45˚ telescopes. In dynamic video-endoscopy, the degree and duration of soft palate closure, and hesitant, incomplete or split/multiple swallows, posterior rotation of the posterior lamina of the ET cartilage during swallowing, any adenoid or lymphoid tissue-related impingement of this rotation, presence and degree of ET orifice widening, any evidence of dynamic ET orifice narrowing, and presence and timing of mTVP contraction are assessed. All these assessments are critical in differentiating the ETD endotypes, i.e., ETD-I, ETD-R, ETD-M.

## Characteristics of the different ET Endotypes

**Normal eustachian tube.** (ETF-N) An ear with normal ETF is expected to have no significant history of OM or OME, especially after age 3. On examination, a clear ME, neutral TM position with good mobility on pneumatic otoscopy and a Type A tympanogram are expected. Sensations of ME fullness/pressure should be relieved by "popping" of the ear with swallowing, yawning or mandibular movements without much difficulty. Under direct visualization, the TM should not present synchronous movement with breathing or retraction with sniffing. After a forceful Valsalva with ET opening, an outward displacement of the TM is expected to be maintained until a maneuver equilibrates the increased MEP, returning the TM to its neutral position. An ear with a non-intact TM (with a VT or a perforation) may also have normal ETF, and when tested is expected to have most test results within the ranges in Table 2.

**Patulous Eustachian Tube Dysfunction (ETD-P).** Patients with a constantly patent ET lumen suffer from hearing the constant noises of breathing, talking and eating (autophony), perceived as louder than normal due to direct transmission of sounds generated at the oropharynx and NPx. There are concurrent variations in MEP synchronous with the changes in NP pressure. Changes may be within the normal range of the breathing cycle or abnormally high with nose blowing or low with sniffing. It is not uncommon for these patients to report a history of recurrent AOM and eventually OME. In a non-intact TM, the OP, RS and CP in the FRT are very low or close to 0 daPa. During Tubomanometry testing, even at the lowest pressure delivery settings, the ear canal probe detects a pressure increase (P1) concurrent with the nasal pressure increase (C1). In a non-intact TM, the ear canal pressure curve and magnitude mirror the nasal pressure curve. Semi-patulous Eustachian Tube Dysfunction (ETD-SP) Ears with semi-patulous ETD may be patulous at times and therefore may present with complaints consistent with ETD-P. On otoscopy, it may be possible to see sudden TM outward or inward displacement, as the NP pressure exceeds the OP. This would more likely occur when there is more forced breathing and contralateral pinched nostril.

**Eustachian Tube Dysfunction with Stricture (ETD-S).** A history of various forms of OM including RAOM, COME, TM-R/RP, ME atelectasis, cholesteatoma, multiple sets of tubes, and prior failed tympanoplasty attempts, brings the question of a structural abnormality, i.e., inability to overcome tissue pressures and difficulty opening the ET. However, having sequelae (including TM-R/RP or atelectasis) or complications does not indicate ETD-S, or even ongoing ETD at the time of evaluation. Pneumatic otoscopy or pneumatic otomicroscopy is useful in assessing the severity of negative MEP. If the TM-R/RP is just a sequela, even a retracted TM can be easily displaced in and out with pneumatic otoscopy. ME adhesions may affect this impression and the degree of displacement of non-attached sections of the TM typically gives an impression as to the degree of negative MEP. Tympanometric curve type B or C is suggestive of ETD-S and a type A tympanogram, especially if obtained at multiple different

**Table 2. Specific findings and test results and suggested ETD endotypes.**

| ASSESSMENT / TEST | Condition / Finding | EUSTACHIAN TUBE FUNCTION ENDOTYPE | | | | | | |
|---|---|---|---|---|---|---|---|---|
| | | Normal | ETD-S | ETD-R | ETD-M | ETD-I | ETD-A | ETD-P /SP |
| **Otoscopy** | **Intact TM** | Neutral | Retr-OME | Retr-OME | Retr-OME | Retr-OME | Retr-OME | Neutral# |
| | **Non-intact TM** | | | | | | | |
| **Tympanometry** | **Intact TM** | Type A | Type C-B | Type C-B | Type C-B | Type C-B | Type C-B | Type A |
| | **Non-intact TM** | NA | NA | NA | NA | NA | NA | NA |
| **Trans Nasal Video-Endoscopy** | **Inflammation** | N | N | N | N | A | N | N |
| | **Lymphoid-Adenoid** | N | A | N | N | N | N | N |
| | **Soft Palate Elevation** | N | N | N | A | N | N | N |
| | **Posterior Rotation** | N | N | A | A | N | N | N |
| | **ET NP Orifice Widening** | N | A | A | A | A | N | N |
| **Sonotubometry** | **SPL increase** | Y | N | N | N | N | N | Cont. |
| **Forced Response Test/ non-intact TM Pressure (daPa)** | **Opening pressure (OP)** | 200–500 | >500 | >500 | 200–500 | >500 | >500 | <200 |
| | **Steady resistance (RS)** | <14 | ≥14 | <14 | <14 | >14 | <14 | <14 |
| | **Closing Pressure (CP)** | 30–120 | >120 | 30–120 | 30–120 | >120 | 30–120 | <30 |
| | **Dilatory Efficiency (DE)** | >1 | >1 | ≤1 | 1 | >1 | >1 | >1 |
| **Pressure Equilibration Test with Inflation-Deflation Test (IDT) or Pressure Chamber (PrCh)** | **%MEPEq-Sw1/+200** | >40% | <40% | <40% | <40% | <40% | <40% | 100 |
| | **%MEPEq-Sw5/+200** | >80% | <80% | <80% | <80% | <80% | <80% | 100 |
| | **%MEPEq-Sw1/-200** | >20% | <20% | <20% | <20% | <20% | <20% | 100 |
| | **%MEPEq-Sw5/-200** | >60% | <60% | <60% | <60% | <60% | <60% | 100 |
| **Opening with Tubo-manometry** | **30 mbar** | Y | N | N | N? | N | N | Y |
| | **40 mbar** | Y | N | N | N? | N | N | Y |
| | **50 mbar** | Y | N? | Y? | N? | N? | Y? | Y |
| **Opening with Valsalva** | **Low pressure (LP)** | N | N | N | N | N | N | Y |
| | **High pressure (HP)** | >10% | <10% | <10% | <10% | <10% | <10% | >50% |
| | **Residual pressure (RP)** | L | H | N | N | H | N | L/none |
| | **RP Clearance (RP-C)** | Y | N | N? | A | A | A | Spont. |
| **Opening with Sniffing** | **Low pressure (LP)** | N | N | N | N | N | N | Y |
| | **High pressure (HP)** | Y? | N | N? | Y? | N | N | Y |
| | **Residual pressure (RP)** | L | | H | N | H | N | Low |
| | **RP Clearance (RP-C)** | Y | | A | A | A | A | Spont. |

(*Continued*)

**Table 2.** (Continued)

| ASSESSMENT / TEST | Condition / Finding | EUSTACHIAN TUBE FUNCTION ENDOTYPE | | | | | | |
|---|---|---|---|---|---|---|---|---|
| | | Normal | ETD-S | ETD-R | ETD-M | ETD-I | ETD-A | ETD-P /SP |
| **Potential Treatment Option(s)** | | None | BDET | Removal of tissue | Soft Palate Exercises | Treat the cause +/- BDET | Surfactant / Mucolytic | Restrictive options* |

Highlighted sections = Abnormal results.

Abbreviations = N: Normal; Y: Yes, present; A: Abnormal; NA: Not applicable; ETD: Eustachian tube dysfunction; ETD-P: Patulous; ETD-SP: Semi-patulous; ETD-S: Obstructive -Stricture; ETD-I: Obstructive-inflammatory; ETD-R: Obstructive- Restrictive /Lymphoid/ Adenoid tissue; ETD-A: Obstructive–Adhesive; ETD-M: Active muscular; TM: tympanic membrane; BDET: Balloon dilation of Eustachian Tube; %Corr-Sw1-5: Percent correction of pressure gradient with 1 and 5 swallows; Nor: Normal; Min: Minimal; Spont.: Spontaneous; Adx: Adenoids; Lymph: Lymphoid; SPL: Sound pressure level.

#: Tympanic membrane movement synchronous with breathing, pronounced with forceful breathing while occluding contralateral nostril.

*: No established effective treatment available.

instances (to exclude a recent incidental or intentional ET opening) should practically rule out ETD-S. Inability to easily open the ET results in rare openings on sonotubometry, high OP, RS, and CP, low %MEPEq (even for equilibration of +MEP) and an absent R value on tubomanometry tested at 30 and 40 mbar (possible present at 50 mbar). Tympanometry after Valsalva and tubomanometry is expected to reveal high residual pressures.

**Active muscular Eustachian Tube Dysfunction (ETD-M).** An ear with a normal TM and clear ME does not imply the presence of normal muscular function sufficient to actively dilate the ET lumen. Soft palate movement by itself involves the mLVP and even a little mTVP activity. Therefore, if there is soft palate movement during swallowing, eating or talking, the position of the palate reflects the changes in the vectors of the muscles and anatomical relationships at and around the nasopharyngeal end and even the entire cartilaginous ET. At rest, the round mLVP bundle sits almost immediately below the ET lumen, and antero-inferiorly to the posterior lamina of the ET cartilage. With contraction of the mLVP sling—muscle bodies from each side—there is a postero-superior displacement of the muscle bundles moving the soft palate to meet the posterior wall of the pharynx, closing the velum and rotating the axis of the ET posterior lamina posteriorly. The degree of rotation determines the degree of widening of the ET pharyngeal opening, which in turn contributes to an effective opening of the ET lumen during mTVP contraction. The main ETF characteristics of ETD-M are normal passive properties with poor %MEPEq.

**Restrictive Eustachian Tube Dysfunction (ETD-R).** The association between adenoid tissue and OM/ETD, including residual or regrowth of adenoid tissue after adenoidectomy, has already been established [32–34]. Possibly also stimulated by other inflammatory causes, the adenoid tissue and lymphoid tissue on the lateral wall of the NPx often extend over the ET cartilage mucosa and sometimes onto the ET orifice. In addition to the static bulk of lymphoid tissue that surrounds the ET orifice, pharyngeal adenoid tissue may also directly impede the rotation of the ET posterior lamina even during an adequate soft palate elevation. As illustrated in Fig 1, when the adenoid tissue is bulky, the soft palate elevation pushes the lymphoid tissue over and onto the ET orifice, further obstructing the ET orifice. This dual mechanism of action may manifest with both abnormal passive and active values on ETF tests that mimic the characteristics of stricture (ETD-S) and poor active muscular function (ETD-M) endotypes. It is possible that the FRT swallow "constriction" phenomenon—reduction of airflow rather than an increase—during the steady flow phase may be related to this protrusion and blockage of the ET pharyngeal opening. Knowledge of this ETD mechanism especially in older children

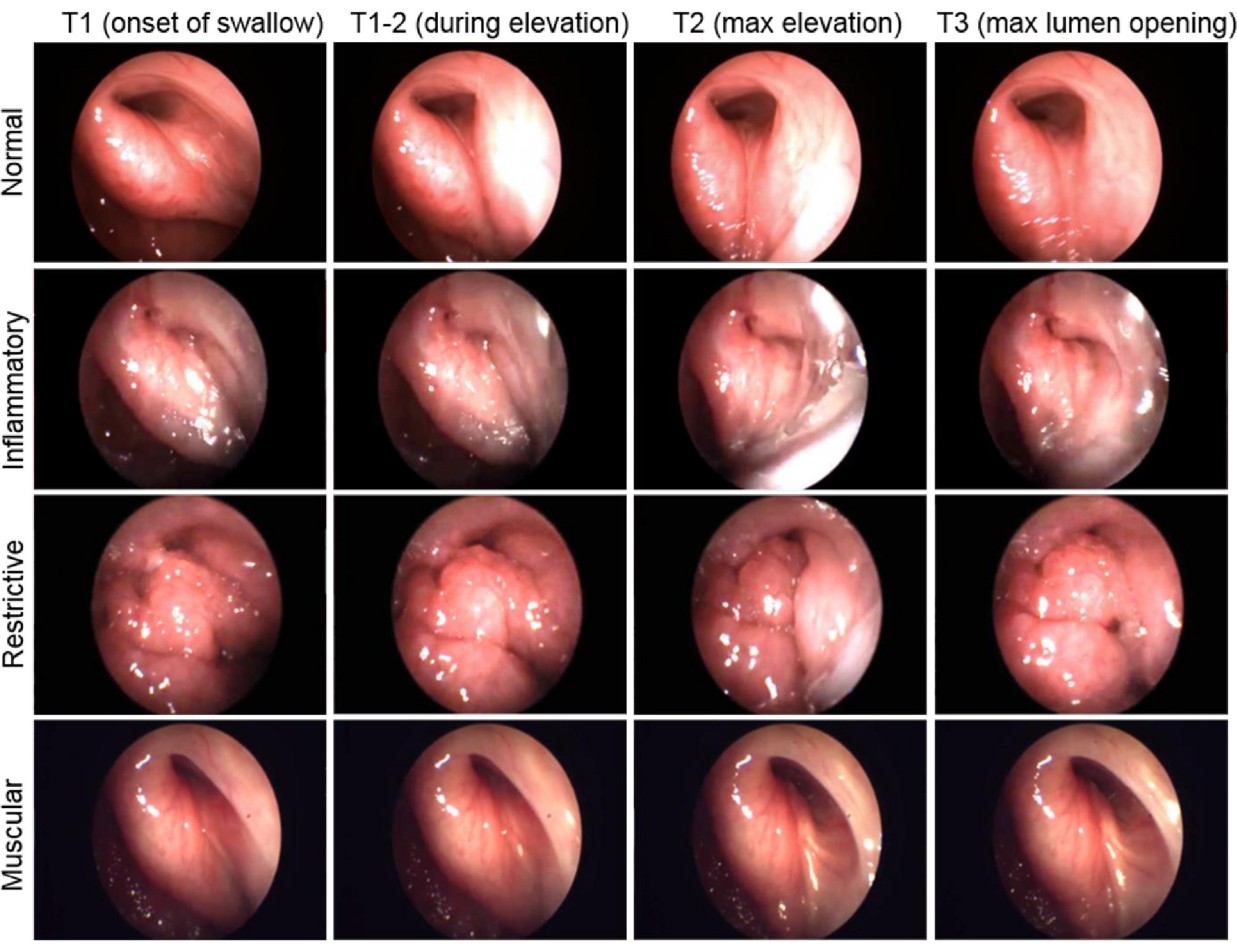

**Fig 1. Demonstrate appearance of the left Eustachian tube nasopharyngeal orifice in four different patients that represent the endotypes of normal Eustachian tube (ET) function, Inflammatory ET dysfunction (ETD), Restrictive ETD, and muscular ETD, at different phases of swallow, i.e., soft palate position elevation at rest or onset of swallow at time-point 1 (T1), during the elevation of soft palate (T1-2), at the time of maximum elevation (T2), and at the time of maximum lumen ET elevation.**

and young adults, suggests including endoscopic evaluation of peri-tubal tissues at rest and during swallowing to verify their role and to have a staged approach to management. Revision adenoidectomy is expected to result in improvement of both passive and active ETD.

**Inflammatory Eustachian Tube Dysfunction (ETD-I).** Inflammation in the NPx, ET orifice and/or lumen may lead to ETD that manifests itself by symptoms and findings similar to ETD-S. ETD-I is characterized by inflammation arising from chronic causes such as nasal allergies, chronic rhinosinusitis or GERD. In addition to test results that are consistent with ETD-S, nasopharyngeal endoscopy is crucial in detecting the increased secretions over the ET orifice and inflamed mucosa over the Torus Tubarius and into the ET orifice. Viral or bacterial acute upper respiratory tract infections may present as temporary ETD-S but are expected to be of short duration. Complete resolution of ETD, even though rarely achieved, is expected with full control of the cause of inflammation.

**Adhesive Eustachian Tube Dysfunction (ETD-A).** Opening of the ET lumen is facilitated by a pressure gradient between the ME and the NPx and dilatory muscular activity that overcome the forces that keep the lumen closed. Three distinct factors impact the separation of

ET lumen surfaces as it dilates: 1) tissue forces and elasticity that keep the lumen closed, 2) presence and effectiveness of inherent surface tension-lowering substances that facilitate separating the opposing surfaces of the ET, 3) molecular forces that the mucous layer exerts to keep the lumen closed. The effects of the latter two factors manifest themselves as adhesive ETD (ETD-A). Although not a true adhesion, lack of sufficient natural surfactant and abundance of molecular bonds formed between the mucous layers of the opposing surfaces work against easy opening of the ET lumen. On FRT, these mechanisms cause initial difficulty in opening the ET lumen (high OP) but normal or even low RS and CP, which differentiates ETD-A from ETD-S. ETD-A also manifests itself as easier openings when FRT is repeated.

Table 2 lists the parameters and ranges used to identify each ETD endotype (abnormal findings/ results/ ranges are highlighted). A question mark was assigned when a specific range could not be well defined, indicating that more research for that parameter is still needed.

## Results

Video-endoscopic and ETF test results were obtained for 71 ears of 40 subjects (22 males, 18 females; 38 white, 2 black), with an average age of 22.9 ± 16.5 years (min: 6.2, max: 64.1).

Distribution of ETD endotypes in the studied population is shown in Table 3. Some ears had multiple endotype assignments. On video-endoscopy 21, 13, 33, 16, 13, NA, NA ETs and on analysis of ETF 20, 24, NA, 38, NA, 3, 13 ears were consistent with ETF-N, ETD-S, ETD-R, ETD-M, ETD-I, ETD-A, and ETD-P/SP, respectively. The zero counts for video-endoscopy and ETF reflect that the specific evaluation modality is not applicable for the assessment of the endotype. Distribution of ears with multiple endotype assignments and the primary endotype are shown on Table 4. Dataset regarding the phenotype assignment is shown in S1 Table.

## Discussion

Research coupled with the clinical testing and care of patients at the UPMC Children's Hospital ETD Clinic has brought us to a critical point in our understanding of ETD. The analysis of examinations and testing results in adults and children validated the roles of active and passive

**Table 3. Distribution of ETD endotypes based on the ETF testing and trans nasal video endoscopy.**

| ETD ENDOTYPE DISTRIBUTION | | | EUSTACHIAN TUBE FUNCTION TEST | | | | | | | Total Number of Endotypes / Ears |
|---|---|---|---|---|---|---|---|---|---|---|
| | | | Normal | ETD-S | ETD-R | ETD-M | ETD-I | ETD-A | ETD-P/SP | |
| | | | 20 | 24 | NA | 38 | NA | 3 | 13 | 98 |
| TRANS NASAL VIDEO ENDOSCOPY | Normal | 21 | 8 | 7 | 0 | 9 | 0 | 1 | 2 | 27 |
| | ETD-S | 13 | 4 | 3 | 0 | 8 | 0 | 0 | 2 | 17 |
| | ETD-R | 33 | 8 | 9 | 0 | 20 | 0 | 1 | 6 | 44 |
| | ETD-M | 16 | 4 | 5 | 0 | 9 | 0 | 1 | 6 | 25 |
| | ETD-I | 13 | 2 | 7 | 0 | 8 | 0 | 0 | 2 | 19 |
| | ETD-A | NA | 0 | 0 | 0 | 0 | 0 | 0 | 0 | 0 |
| | ETD-P/SP | NA | 0 | 0 | 0 | 0 | 0 | 0 | 0 | 0 |
| Total Number of Endotypes / Ears | | | 26 | 31 | 0 | 54 | 0 | 3 | 18 | |
| | | 96 | | | | | | | | 71 |

Abbreviations = NA: Not applicable (not applicable to the assessment of an endotype with the specific modality); ETD: Eustachian tube dysfunction; ETF: Eustachian tube function; ETD-P: Patulous; ETD-SP: Semi-patulous; ETD-S: Obstructive -Stricture; ETD-I: Obstructive-inflammatory; ETD-R: Obstructive- Restrictive /Lymphoid/ Adenoid tissue; ETD-A: Obstructive–Adhesive; ETD-M: Active muscular.

**Table 4. Distribution of multiple and primary ETD endotypes with either trans nasal video endoscopy or ETF testing, or both.** For each ear, a primary, i.e., a dominant diagnosis was selected based on the review of both diagnostic modalities.

| ETD ENDOTYPE | Video endoscopy | ETF | BOTH | PRIMARY |
|---|---|---|---|---|
| Normal | 21 | 20 | 41 | 20 |
| ETD-S | 13 | 24 | 37 | 5 |
| ETD-R | 33 | NA | 33 | 15 |
| ETD-M | 16 | 38 | 54 | 16 |
| ETD-I | 13 | NA | 13 | 8 |
| ETD-A | NA | 3 | 3 | 2 |
| ETD-P/SP | NA | 13 | 13 | 5 |
| TOTAL | 96 | 98 | 194 | 71 |

Abbreviations = NA: Not applicable (not applicable to the assessment of an endotype with the specific modality); ETD: Eustachian tube dysfunction; ETF: Eustachian tube function; ETD-P: Patulous; ETD-SP: Semi-patulous; ETD-S: Obstructive -Stricture; ETD-I: Obstructive-inflammatory; ETD-R: Obstructive- Restrictive /Lymphoid/ Adenoid tissue; ETD-A: Obstructive–Adhesive; ETD-M: Active muscular.

components in ET opening mechanics [35–41]. This work has led to the identification of a constellation of findings and ETF test results, and to the hypothesis of specific ETD endotypes. Endotype implies a distinct mechanism of pathogenesis and/or a set of characteristics that differentiate similar clinical manifestations, as presented in Table 2. While the characteristics and test results that define these distinct features of ETD are not entirely novel, this is a novel approach to establish a better level of understanding of each differentiating feature.

The most comprehensive ETD classification so far was presented by the 2015 Consensus on ETD [42]. Based on the ventilatory function of the ET, they defined three subtypes of ETD: 1) dilatory ETD, 2) baro-challenge-induced ETD, 3) patulous ETD. Dilatory ETD was further subclassified as: 1a) functional obstruction, 1b) dynamic dysfunction (muscular failure), and 3) anatomical obstruction. The authors further introduced a time-based definition as acute dilatory ETD, and chronic dilatory ETD, and summarized their classification as acute ETD, chronic ETD, baro-challenge-induced ETD and patulous ET. This classification took into consideration specific symptoms, presentation, tympanograms and a few classical findings, such as TM excursion concurrent with breathing for the patulous subtype. Although the authors did not use the term, their approach was to define the ETD phenotypes. While the authors mentioned functional, dynamic and anatomical obstruction as subcategories of dilatory ETD, they did not describe these, and gave no further information regarding these in their consensus document. It appeared that this was an attempt to refer to some historical definitions introduced by Bluestone as early as 1974 [43,44]. While such terms in the consensus paper imply a mechanistic approach, this previously well-known concept had already defined mechanical and functional ET obstruction and differentiated anatomical obstruction as extrinsic and intrinsic (intra-luminal) obstruction [43]. Moreover, although anatomical obstruction in the consensus paper appears to cover this previous concept, authors made no clarification regarding functional obstruction and dynamic dysfunction (muscular failure), but introduces more ambiguity.

There are similarities of some of the endotypes described here with the ones of the previous literature including the consensus paper: patulous (ETD-P) and active muscular dysfunction (ETD-M) are comparable to the consensus definitions of Patulous and Dilatory dysfunction—dynamic/muscular. Stricture dysfunction (ETD-S) maps to the consensus criteria for either functional or anatomical dilatory dysfunction but, due to the lack of definition for these subtypes, we were unable to make this determination.

What is novel about our classification presented here is the use of objective measures to assess the ETF active and passive parameters, which permits subcategorization of ETD obstruction to stricture (ETD-S), inflammatory (ETD-I), restrictive (ETD-R) and adhesive (ETD-A) endotypes. Endoscopic evaluation alone is unable to diagnose ETD-A, ETD-S or ETD-P, while ETF testing alone cannot diagnose ETD-R or ETD-I. Therefore, both diagnostic modalities are required to differentiate the full spectrum of ETD endotypes.

As seen in Table 3, in 71 ears, a total of 96 endotype assignments were made with video-endoscopy, and 98 endotype assignments with ETF testing. Overlap in categories assigned with video-endoscopy was due to concurrent presence of adenoid tissue, inflammation and/or inadequate soft palate elevation. Overlap in categories with ETF testing was mostly due to assignments of endotypes based on the passive and active muscular features, since in some subjects both were abnormal. While ETF test results were normal in 20 ears, video-endoscopy was normal in only 8 of those, implying abnormal endoscopic appearance would have falsely assigned ETD-S in 4, ETD-R in 8, ETD-M in 4 and ETD-I in 2 ears. Similarly, despite normal video-endoscopy, 7 ears had ETD-S, 9 ears had ETD-M, 1 ear had ETD-A, and 2 has ETD-P, highlighting the importance of accurate and detailed ETF tests to determine ETD. Upon the establishment and validation of clear descriptions of ETD endotypes, it will be possible to assign primary and secondary categorizations. Based on the existing data and suggested criteria, a primary endotype was assigned to each of 71 ears in this dataset (Table 4).

Although not entirely novel, description and differentiation of ETD endotypes will bring attention to the highlighted characteristics and test results to diagnose ETD in a patient. Distinguishing the mechanisms of ETD for each endotype dictates a specific corrective action that is a targeted treatment modality. It is obvious that the recently popularized balloon dilation procedure in some contexts will not only be ineffective but would be contra-indicated in ETD-P or ETD-SP. Apparently ETD-R may mimic ETD-M, and removal of adenoid and lymphoid tissues may resolve the ETD. Even though ETD-I may be targeted with balloon dilation and may improve, especially as per the evidence on very small sample sizes obtained by different subjects pre- and post- balloon dilation [24], treating the underlying inflammatory causes is essential, as inflammatory changes in the lumen could return if the risk factors persist. In the absence of structural or other functional problems, the ETD-A endotype may be suitable for treatment with mucolytic or surface tension lowering agents.

Identification of an ETD endotype does not necessarily imply a directly corresponding treatment modality. Only after performing adenoidectomy and removal or peri-tubal lymphoid tissues to eliminate the mechanical restriction in the ETD-R endotype can the residual ETD-M component be identified. Although BDET could be offered to patients with ETD-M to lower the necessary effort to overcome the tissue pressures and to achieve the threshold to open the ET, a potential alternative approach could include strengthening the peri-tubal muscles. Further studies should investigate the role of exercises that strengthen and prolong mLVP contraction, therefore enhancing velar closure and posterior rotation of the posterior lamina of the ET cartilage and widening of the ET orifice, such as exercises using the Expiratory Muscle Strength Training (EMST) device (Aspire Respiratory Products, Cedar Point, NC). Similarly, ETD-S does not necessarily directly imply the need for BDET. History, duration of symptoms, laterality and other findings should be considered before making that recommendation, as this endotype includes extrinsic and intrinsic anatomical and mechanical causes, all manifesting themselves as similar in the testing and evaluation work-up. Even in the absence of any apparent mechanical cause on endoscopy, elements in the history such as unilateral new or recent onset without prior history of OM or otologic symptoms may direct the clinician to obtain imaging to rule out a mass that is not apparent on visualization.

As acceptance of common terminology and criteria for identifying and differentiating ETD endotypes occurs, trials to test and validate these proposed endotypes with targeted specific treatment methods should follow. This attempt to describe ETD endotypes is in no way a claim to have a complete and final set of definitions and classifications of ETD or have a certain final set of criteria for each of these endotypes. Rather, this introduces a paradigm to approach, understand and test for ETD, and for refining the specifics of the endotypes through the collective effort of multiple centers. Unfortunately, the current ETF test battery is not available in most centers that are currently assessing, diagnosing and treating ETD. While differentiation and validation of ETD endotypes may require more sophisticated tests, the ultimate goal is to develop relatively simple test methods that still have high diagnostic accuracy so that similar standardized tests can be utilized for diagnosis and monitoring of outcomes in clinical centers.

## Supporting information

**S1 Table.**
(XLSX)

## Author Contributions

**Conceptualization:** Cuneyt M. Alper.

**Data curation:** Cuneyt M. Alper, J. Douglas Swarts.

**Formal analysis:** Cuneyt M. Alper, J. Douglas Swarts.

**Funding acquisition:** Cuneyt M. Alper.

**Investigation:** Cuneyt M. Alper, J. Douglas Swarts.

**Methodology:** Cuneyt M. Alper, J. Douglas Swarts.

**Project administration:** Cuneyt M. Alper, Miriam S. Teixeira.

**Resources:** Miriam S. Teixeira, Ellen M. Mandel, J. Douglas Swarts.

**Software:** J. Douglas Swarts.

**Supervision:** Cuneyt M. Alper, Miriam S. Teixeira.

**Validation:** Cuneyt M. Alper, J. Douglas Swarts.

**Writing – original draft:** Cuneyt M. Alper.

**Writing – review & editing:** Miriam S. Teixeira, Ellen M. Mandel, J. Douglas Swarts.

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
