## [Decision Letter · Decision Letter 0]

21 Mar 2023

Dissecting Eustachian Tube Dysfunction: From Phenotypes to Endotypes

PONE-D-23-01368

Dear Dr. Alper,

We’re pleased to inform you that your manuscript has been judged scientifically suitable for publication and will be formally accepted for publication once it meets all outstanding technical requirements.

Kind regards,

Jorge Spratley, MD, PhD

Academic Editor

PLOS ONE

Additional Editor Comments (optional):

Congratulations for the excellent manuscript. This paper fills an important gap in the knowledge of the Auditory Tube pathophysiology.

Reviewers' comments:

Reviewer's Responses to Questions

**Comments to the Author**

1. Is the manuscript technically sound, and do the data support the conclusions?

Reviewer #1: Yes

Reviewer #2: Yes

2. Has the statistical analysis been performed appropriately and rigorously? 

Reviewer #1: Yes

Reviewer #2: N/A

3. Have the authors made all data underlying the findings in their manuscript fully available?

Reviewer #1: Yes

Reviewer #2: Yes

4. Is the manuscript presented in an intelligible fashion and written in standard English?

Reviewer #1: Yes

Reviewer #2: Yes

5. Review Comments to the Author

Reviewer #1: This manuscript corresponds to the publication criteria of PLOS ONE. It is original in that it proposes a clinical and para-clinical classification, accurate and nuanced, while will help and optimize the diagnosis and guide towards a focused and more efficient therapeutic management of ETD. This terminological approach and criteria for differentiating EDT end-types also allows for better exploitation of research results from multiple centers.

Reviewer #2: The manuscript entitled “Dissecting Eustachian Tube Dysfunction: From Phenotypes to Endotypes” presents a large and exhaustive analysis of eustachian tube disfunction (ETD) patients. As patient-reported outcome measures have been shown not to present a clear diagnostic value the study presented aimed to analyze in an objective manner the possible causes of ETD and the distribution of specific endotypes in the studied population. These diverse endotypes suggests a different mechanism of pathogenesis. The is a relevant and well written study which concludes that ETD should be diagnosed on the basis of clinical assessment and tests of ET function and the selection of the best treatment should be adjusted for each endotype or group of coexisting endotypes.

6. PLOS authors have the option to publish the peer review history of their article (what does this mean?). If published, this will include your full peer review and any attached files.

Reviewer #1: **Yes: **Ars Bernard

Reviewer #2: No

---

## [Editor Report · Acceptance letter]

11 Apr 2023

PONE-D-23-01368 

Dissecting Eustachian Tube Dysfunction: From Phenotypes to Endotypes 

Dear Dr. Alper:

I'm pleased to inform you that your manuscript has been deemed suitable for publication in PLOS ONE. Congratulations! Your manuscript is now with our production department. 

Kind regards, 

on behalf of

Professor Jorge Spratley 

Academic Editor

PLOS ONE